# Impact of Gluten-Free Sorghum Bread Genotypes on Glycemic and Antioxidant Responses in Healthy Adults

**DOI:** 10.3390/foods10102256

**Published:** 2021-09-23

**Authors:** Lorenza Rodrigues dos Reis Gallo, Caio Eduardo Gonçalves Reis, Márcio Antônio Mendonça, Vera Sônia Nunes da Silva, Maria Teresa Bertoldo Pacheco, Raquel Braz Assunção Botelho

**Affiliations:** 1Department of Nutrition, College of Health Sciences, University of Brasilia, Brasilia 70910-900, Brazil; caioedureis@gmail.com; 2College of Agronomy and Veterinary Medicine, University of Brasilia, Brasilia 70910-900, Brazil; marcioamen@gmail.com; 3Institute of Food Technology, Secretariat of Agriculture and Supply of Sao Paulo, Sao Paulo 13070-178, Brazil; vera.silva@ital.sp.gov.br (V.S.N.d.S.); mtb@ital.sp.gov.br (M.T.B.P.)

**Keywords:** gluten-free, sorghum, bread, antioxidant activity, resistant starch, dietary fibers

## Abstract

Sorghum is used to provide good quality gluten-free products due to phytochemicals and low glycemic index (GI). This study aimed to determine the chemical composition, the antioxidant activity and capacity, and the glycemic and insulinemic responses of gluten-free (GF) sorghum bread. GF bread samples were produced with three different sorghum genotypes. The samples were analyzed for chemical composition, resistant starch and dietary fiber content; antioxidant activity by ORAC; antioxidant capacity by FRAP; GI; and insulinemic responses. This double-blind, crossover, randomized clinical trial was conducted with 10 healthy men aged 28.0 ± 4.9 years (77.6 ± 11.7 kg and 24.2 ± 2.3 kg/m^2)^. All sorghum bread showed significantly more fiber than rice bread (control). Brown sorghum bread was classified as low GI, bronze and white as medium GI, and control as high GI. Brown sorghum bread presented a low carbohydrate content, a significant amount of fiber, and a significantly lower 3 h AUC glucose response than those of the control, aside from the highest antioxidant activity value (*p* ≤ 0.001). Therefore, brown sorghum was superior to other genotypes analyzed in this study, and its production should be encouraged to provide gluten-free products with a better nutritional profile. More research is required to explore the effects of different sorghum genotypes in food products on human health.

## 1. Introduction

There is a growing demand for gluten-free products (GFP) due to adverse symptoms of gluten such as celiac disease, gluten or wheat allergy, and gluten sensitivity [1,2]. Many wheat flour substitutes are applied to produce GFP, including sorghum (*Sorghum bicolor L.*) [3], a gluten-free (GF) grain with a simple cultivation process, high nutritional profile, and health benefits [4,5,6,7]. Sorghum can be incorporated into gluten-free bread formulations, as well as other baked products, such as cakes and cookies, flakes, and pasta [8,9,10,11,12].

Several sorghum genotypes are presented by the genotypes white, cream, yellow, orange, bronze, red, brown, black, and various combinations of these colors [13]. Each genotype presents different characteristics. For example, a white-colored genotype with no pigmented testa presents no significant levels of tannins; a red-colored genotype with pigmented testa presents a moderate tannins content; and a brown-colored genotype with pigmented testa presents a high tannins content [14].

Phenolic compounds of sorghum are usually concentrated in the grain’s pericarp [15], and sorghum genotypes containing tannins have a higher antioxidant capacity than sorghum genotypes that do not contain tannins [8]. Moreover, in comparison with white sorghum genotypes, colored sorghum genotypes have a higher phenolic compounds concentration [16]. Therefore, brown-, bronze- and red-pigmented sorghums are rich in phenolic compounds (e.g., flavonoids). They are also rich sources of several phytochemicals, including tannins, phenolic acids, anthocyanins, phytosterols, and policosanols, providing significant antioxidant properties [8,17]. According to Awika and Rooney [8], sorghum genotypes can contain from 0.5 to 68 mg/g of tannins and several different phenolic acids with 3 to 43 mg/100 g of total phenolic content [14].

These sorghum polyphenols are known to function as potent antioxidants, at least in vitro [18]. Despite phenolic compounds’ bioavailability after dietary intake being a research topic in recent years, clinical studies are scarce and controversial [7,19]. According to Prior and Wu [19], for some phenolic compounds, there are differences among their primary forms circulating in blood or tissues after oral ingestion and the original forms in the diet. Moreover, some anthocyanins are absorbed intact, and absorption can be saturated. Additionally, the quantities excreted in the urine are less than 0.1% of the total intake, although 60 to 90% of anthocyanin may disappear from the gastrointestinal tract within 4 h after a meal.

Another relevant characteristic of sorghum is the high level of resistant starch (RS), ranging from 2.2 to 6.5 g/100 g [6]. Moreover, sorghum is fiber-rich, containing over 95% of the non-starch polysaccharides [20]. These characteristics provide a slow starch absorption similar to a low-glycemic index food, positively impacting glucose metabolism [12]. Among several studies regarding sorghum products [8,9,10,11,12], only Wolter et al. [11] have calculated the glycemic index (GI) of one genotype of sorghum in bread using in vitro starch digestibility. This resulted in a medium GI (GI = 69). Therefore, there is a lack of information about sorghum products’ composition and their impact on glucose metabolism. More studies are needed to assess the effects of sorghum bread consumption on human health [11].

Thus, this study aimed to analyze the chemical composition, antioxidant properties, and the effects on the glucose metabolism of GF sorghum bread made with three different genotypes in healthy adults. Additionally, the study focused on evaluating different sorghum genotypes to produce more options for gluten-free products with better nutritional qualities.

## 2. Materials and Methods

### 2.1. Sorghum and Bread Production

According to our group’s previous study, three genotypes of sorghum provided by Embrapa (Brazilian Agriculture Research Enterprise) were chosen by the different pericarp colors (white—BRS 501, brown—BR 305, and bronze—BRS 332) and the highest acceptability [21]. One batch of each sorghum genotype was transformed into flour using a Thermomix processor (speed 10 for 3 minutes) (Vorwek TM6, Wuppertal, Germany).

Four bread samples were prepared according to the recipes used by Andrade de Aguiar et al. [21], one as control with commercial white rice flour (produced with *Oryza sativa*) and three with the different types of sorghum flours (made with white, brown, and bronze genotypes). All formulations had the same proportion of ingredients (22.36% of flour, 10.06% of potato starch, 6.04% of whole egg, 5.36% of soy oil, 4.14% of cassava flour, 3.80% of brown sugar, 3.69% of egg whites, 1.12% of dry yest, 0.56% of salt, 0.39% of xanthan gum, and 42.48% of water), only changing the primary flour (rice or sorghum types) for each type of bread. All ingredients used were gluten-free.

Ingredients were weighed to prepare the four different samples of bread. Dry yeast was mixed with 26% of the total water (T—35 °C) and brown sugar and fermented for 10 min. All the other dry ingredients were mixed in a food mixer for 1 min. Then, egg, egg whites, and soy oil were added and mixed with the dry ingredients for one more minute. Hydrated yeast was added to the previous mixture. Each dough was placed in a rectangular cake tin that measured 8.66 (width), ×3.94 (height), 2.56 (depth) inches for a fermentation process of 25 min then was baked in a gas oven (Brastemp, São Paulo, Brazil). Rice bread dough was baked in a pre-heated oven at 180 °C for 45 min and sorghum bread dough for 50 min. After baking, samples were removed from the tin, and each bread slice was stored in freezer bags. For bread chemical composition, samples were analyzed the day after bread production, and for antioxidant analysis, bread slices were frozen at −80 °C until analysis, approximately for two months. For resistant starch analysis, samples were stored in the same freezer bags at room temperature for 24 h before analysis.

### 2.2. Bread Chemical Composition

For each type of bread, three separate recipes were baked. Then, analyses were conducted in triplicate for moisture (AOAC Official Method 925.09), ash (AACC Official Method 08-03.01), proteins (Kjeldahl, AACC 46-13), and lipids (by extraction with petroleum ether by dragging under pressure with Extractor Ankom Model XT10). Total dietary fiber analysis (AOAC Official Method 985.29) was conducted in dry base samples. Carbohydrates were calculated by difference: 100—(weight in grams (protein + fat + ash + fiber + Resistant Starch (RS)) in 100 g of food). RS was analyzed on wet basis samples by the official analysis methods [22,23,24,25], in which 0.1 g of each sample was incubated in a shaking water bath (Yatherm Scientific, Gautama Buddha Nagar, Uttar Pradesh, India) with 4 mL of pancreatic α-amylase and amyloglucosidase (AMG) for 16 h at 37 °C and 200 strokes/min. The reaction ended with the addition of 4 mL of ethanol (99% *v/v*). The samples were centrifuged (Centrifuge Eppendorf 5702 R) at 3000 rpm for 10 min at 4 °C. The centrifugation pellet was re-suspended in 2 mL of ethanol (50% *v/v*), and then, in 6 mL of ethanol, followed by centrifugation under the same conditions twice. RS in the pellet was dissolved in 2 mL of 2 M KOH by vigorously stirring in an ice-water bath over a magnetic stirrer (Warmnest HJ-3) for 20 min. This solution was neutralized with 8 mL of 1.2 M sodium acetate buffer (pH 3.8), and the starch was quantitatively hydrolyzed to glucose by adding 0.1 mL of AMG. This solution was mixed and incubated for 30 min in a water bath at 50 °C with intermittent mixing. Then, samples were centrifuged at 3000 rpm for 10 min, 0.1 mL aliquots (in duplicate) were transferred into glass tubes, 3.0 mL of glucose oxidase/peroxidase reagent (GOPOD) was added and incubated at 50 °C for 20 min. The last step was to measure the absorbance of each solution in a spectrophotometer (Biochrom, Cambridge, United Kingdom) at 510 nm against the reagent blank (0.1 mL of sodium acetate buffer with pH 4.5 + 3 mL of GOPOD) [22,23,24,25].

### 2.3. Antioxidant Activity

The bread antioxidant activity was determined using an oxygen radical absorbance capacity (ORAC) assay quantified by fluorescence, a standardized method for determining antioxidant capacity in foods [26]. The standard curve was generated using the area under the curve (AUC) for different standard concentrations of Trolox (6-hydroxy-2,5,7,8-tetramethylchroman-2-carboxylic acid). The reagents were phosphate buffer pH 7.4, fluorescein 16,371.10^−8^ mol/L, APPH 178 mmol/L, and Trolox. All extract samples and Trolox standard solutions were pipetted with nine replicates into a black microplate and incubated at 37 °C for 10 min. Then, the peroxyl radical generator 2,2′-azobis (2-amidinopropane) dihydrochloride (AAPH) and the fluorescence were measured at 37 °C every 60 s using a spectrophotometer from 485 and 520 nm. The ORAC values were calculated from the Trolox standard curve with R^2^ = 0.9912 and were expressed as milligrams of Trolox Equivalent (TE) per 1 g of extract (dry weight basis) [27].

### 2.4. Individuals

The sample size calculation was performed using the G*Power software (version 3.1.9.2; Dusseldorf University, Dusseldorf, Germany) [28], assuming glucose levels as the primary variable of the study and based on the result of Poquette et al. [29], with a statistical power of 90% and an alpha error of 5% (two-tails). This resulted in a sample size of 9 participants (crossover design).

Participants were recruited through social media and public advertisements. Eligibility criteria included the following: age 18–50 years; no medications that affect glucose metabolism; slight bodyweight fluctuation (≤5 kg in the past three months); willingness to eat all test foods; no self-reported allergy to the foods provided in the study; no self-reported sleep disorders; no cardiovascular, metabolic, and gastrointestinal diseases; no reported family history of type 2 diabetes mellitus in first-degree relatives; and fasting capillary blood glucose 70–100 mg/dL. Participants with metabolic disorders, who were taking any kind of medicine or allergic to the food provided in the study, or did not conclude all experimental sessions, were excluded.

Ethical approval for the study protocol was obtained from the Ethics Committee of Health Sciences School of the University of Brasilia (CAAE 58257416.1.0000.0030). All participants provided written informed consent before participation.

### 2.5. Clinical Trial Design

This study was a double-blind, crossover, randomized clinical trial in which 13 male individuals were eligible to participate in five experimental sessions with a 3- to 15-day washout period. For screening, participants answered questionnaires based on the recruitment criteria.

All experimental sessions were initiated between 7 and 8 a.m. with participants in 12 h fasting. Participants were instructed not to consume alcohol or perform any non-habitual physical activity 24 h before the sessions. They were also advised to maintain regular dietary intake and physical activity during the study protocol.

At each session, capillary glucose level was assessed by finger stick blood using a glucometer (Accu-check Performa; Roche Diagnostics, Basileia, Suiça) to ensure the food-deprived state (glucose level < 100 mg/dL). Then, an indwelling catheter was placed in the participant’s forearm, and blood samples were drawn at baseline (−10) and 15, 30, 45, 60, 90, 120, and at 180 min after consuming glucose or the test bread. Participants were not allowed to eat or drink anything (except water) besides the test meals provided during the study sessions. They were allowed to read, listen to music, watch TV, use the computer, and use the toilet inside the laboratory.

### 2.6. Anthropometric and Body Composition Measurements

On the first day of the experiment, weight, height, and body fat were measured with 12 h fasting state. Bodyweight was assessed using a weight scale with 150 kg capacity and 50 g precision (OMRON model HN-289) and a stadiometer for height measuring to the nearest 0.1 cm (Wiso model Series 12). Body mass index (BMI) was computed and classified according to WHO [30]. Body fat percentage was measured by tetrapolar bioelectrical impedance (Body Compositon Analyzer—Quantum II, RJL Systems) according to the protocol described by Lukaski et al. [31].

### 2.7. Test Meals

Five experiment sessions were performed with anhydrous glucose or four bread samples made with rice flour (control) or white, brown, and bronze sorghum, all containing 50 g of available carbohydrate. Test meals were chosen to compare three different genotypes of sorghum and control bread. In each session, the participant consumed the test meal within 10 min in a randomized order. The simple randomization process was performed using a random number table [32]. The researchers were blinded to all experimental treatments, and the participants were blind to the type of bread consumed.

### 2.8. Antioxidant Capacity

To determine the blood antioxidant capacity, 3 mL of blood was collected in EDTA tubes with GSH 0.65 mmol/L and centrifuged at 4000 rpm for 15 min at 4 °C. The plasma was frozen in liquid nitrogen and stored at −80 °C until analyses. Antioxidant capacity analyses were conducted by FRAP (Ferric Reducing Ability of Plasma) method [33].

### 2.9. Biochemical Measurements

A volume of 4 mL of blood was collected in a red top vacutainer at each time point. After clotting and centrifugation at 4000 rpm for 15 min at 4 °C (Centrifuge Eppendorf 5702 R), glucose and insulin concentrations were measured by glucose oxidase (ADVIA, model 2400, Siemens Healthcare Diagnostics S.A., São Paulo, Brazil) and electrochemiluminescence methods (ADVIA, model Centaur, Siemens Healthcare Diagnostics S.A., São Paulo, Brazil), respectively. Sensitivity of glucose oxidase was 0.12 mmol/l (within-run CV of 0.41%) and insulin immunoassay was 1.39 pmol/l (within-run CV of 1.9%). The incremental AUC for glucose and insulin (3 h) was calculated, excluding the values below the baseline values, based on the trapezoidal method [31] using Microsoft Excel software, version 2010 (Microsoft Corporation, Washington, USA). The glycemic (GI) and insulinemic index (II) of the sorghum bread samples were determined based on the 2 h AUC response compared with glucose response as standard value (100) [34].

### 2.10. Statistical Analysis

Levene’s and the Shapiro–Wilk tests were performed to determine data homogeneity of variance and normality, respectively. One-way analyses of variance (ANOVA) with Tukey post hoc test was applied to assess food composition, antioxidant status (ORAC and FRAP), glucose, and insulin AUC between glucose and the test meals (control and sorghum bread—white, brown, and bronze). Two-way repeated-measures ANOVA was used to examine the effects of test meals on postprandial glycemic and insulinemic responses with Bonferroni adjustments as post hoc comparisons when significant meal versus time interactions were found. The effect size of glucose and insulin responses and glucose and insulin AUC were calculated using ɳ^2^ Eta squared. Statistical analyses were conducted using Statistical Package for the Social Sciences, version 21 (SAS Institute, Inc., Cary, North Carolina, USA). Differences were considered significant at *p* < 0.05 (two-tailed). The results are presented as mean values and standard deviations.

## 3. Results

### 3.1. Individuals Characteristics

Of 13 individuals initially recruited, two did not conclude all the phases, and one was excluded due to a medical condition. The excluded participant had a basal insulin and insulin response higher than the healthy participants. Therefore, data of ten healthy non-celiac males were analyzed. The ten participants that concluded the experiment presented 28.0 ± 4.9 years, 77.6 ± 11.7 kg, 1.78 ± 0.07 m, 24.2 ± 2.3 kg/m^2^, 21.36% of body fat, and capillary fasting blood glucose of 89 ± 4.3 mg/dL. All participants were non-smokers.

### 3.2. Bread Chemical Composition

As shown in Table 1, the control bread presented a statistical difference in moisture compared to sorghum bread samples (*p* < 0.05), and white sorghum bread showed significantly less ashes compared to other samples (*p* < 0.05). For carbohydrates, all samples of sorghum bread presented less carbohydrate content compared to control (rice bread) (*p* < 0.05). Additionally, brown bread presented significantly more carbohydrates compared to bronze (*p* = 0.001). For RS, control and bronze bread samples presented higher values compared to the other sorghum genotypes (*p* < 0.05) with no statistical difference between bronze and control (*p* = 0.16). For fiber, all sorghum bread presented significantly higher values compared with control (*p* < 0.05), with brown presenting the highest content, followed by white and bronze bread (all *p* < 0.05). Bronze bread presented a higher value of protein and lipids than the other bread samples, with a significant difference among all samples for lipids content (*p* < 0.05).

Brown sorghum bread presented significantly higher ORAC antioxidant activity compared to the others (*p* ≤ 0.001). Moreover, white and bronze sorghum showed no significant difference from the control bread (*p* = 0.38 and *p* = 0.09, respectively) (Table 1).

### 3.3. Postprandial Glucose and Insulin Responses

As presented in Table 2, regarding 3 h AUC glucose response, brown bread presented a lower value than the control bread and glucose drink (*p* < 0.05; ɳ^2^ = 0.132). Additionally, there were no significant differences among sorghum bread samples. For 3 h insulin AUC, there were no statistical differences among all bread samples. The glucose drink AUC response was significantly higher than other bread samples (*p* < 0.05).

According to the GI classification (high GI ≥ 70; intermediate GI 56–69; low GI ≤ 55)^30^, brown sorghum bread presented a low GI, bronze, and white sorghum an intermediate GI, and a high GI was found only for the control bread. For the insulin index, regarding the bread samples, the control showed a higher score (70) compared to sorghum bread (56, 50 and 44 for bronze, brown and white genotypes, respectively) (Table 2).

There were no significant time versus meal interaction effects at any time point on glycemic and insulinemic responses between tested meals (control and sorghum bread) (*p* ≥ 0.64 and *p* ≥ 0.48, respectively) (Figure 1).

### 3.4. Antioxidant Capacity

There was no significant time versus meal effect on blood antioxidant capacity (FRAP) between tested meals (control and sorghum bread samples) with *p* ≥ 0.504.

## 4. Discussion

Based on the previous beneficial effects of sorghum bread intake on glucose metabolism [16], our findings support this hypothesis, showing improved glycemic and insulinemic responses after consuming sorghum bread in healthy adult men. All sorghum bread samples presented lower GI and II when compared to the control (rice bread). This is important because it was possible to diminish the interaction effects among the ingredients (flours, starch, eggs, and xanthan, etc.) since all the GF bread samples had precisely the same proportion of potato starch, cassava flour, and xanthan gum. Additionally, sorghum bread samples presented similar moisture contents, with no statistical difference. However, the control bread showed significantly less moisture (*p* < 0.05) compared to sorghum samples.

Several studies have found differences in chemical composition among sorghum genotypes such as tannins, phenolic compounds, resistant starch, and fiber content [7,8,14,17,19]. Regarding the sorghum bread genotypes, the brown sorghum bread presented lower carbohydrate amounts, lower lipids, and significantly higher fiber content and antioxidant activity (ORAC) than other sorghum bread genotypes. White sorghum bread showed low carbohydrate and lipids content and a high fiber amount. In contrast, bronze sorghum bread presented a higher resistant starch and protein content than other sorghum bread genotypes.

Brown sorghum bread showed a significantly lower 3 h AUC glucose response than the control bread and presented a low GI value. Since low GI is related to better glycemic response, brown sorghum bread can be considered a good alternative to improve glycemic response. According to Westman et al. [35], a diet with a low GI improves hemoglobin A1c, fasting glucose, and insulin compared to a normoglycemic diet in individuals with obesity and type 2 diabetes.

In our study, it was possible to determine that all sorghum bread samples presented a significantly higher fiber content than control bread (*p* < 0.05), contributing to the lower GI classification compared to the control. However, white sorghum bread presented the highest GI among sorghum bread samples. Its low resistant starch content can explain its low lipids and protein content, confirmed by Al Dhaheri et al. [36]. According to this study, which investigates the effect of nutritional composition on the glycemic index of different foods, samples with a high protein content reduce the glycemic response. Additionally, food’s high fat and fiber content was related to a decreased postprandial glycemic response [36]. Therefore, brown sorghum bread has better nutritional effects since it shows significantly higher fiber and antioxidant activity, presenting the lowest GI.

Besides the fiber content, predominantly brown and white sorghum bread, all samples are GF, since gluten is not part of any ingredient and none presented cross-contamination industry disclosure. It is important since GFP tend to present less fiber content because they are usually prepared with corn starch, potato flour/starch, and low-fiber rice. Rice flour is one of the most used to produce GFP; however, it presents a low protein content and quickly digested carbohydrates [37].

According to Calvo-Lerma et al. [38], most GFP has a low nutritional profile, especially in bread and pasta. Similarly, Melini and Melini [39] who reviewed nutritional profile of GFP available on the market, showed that several GFP have a low protein and high fat and salt content compared to their equivalent gluten-containing products. Additionally, rice and corn that are the most frequently used in formulation of GFP are lacking in protein and fiber [40]. Other studies with GFP showed that white rice flour presented only 2.4 g of total fiber per 100 g, GF bread made with corn and potato starch had 3.34 g per 100 g, and commercial GF bread presented 1.2 to 5.6 g per 100 g of fiber [37,38,41,42]. On the other hand, the present study presented a better food composition for sorghum bread than rice flour bread (control). Similarly, Hariprasanna et al. [43] demonstrated that sorghum grain had a better nutritional profile than rice.

All sorghum bread samples presented more than 4.5 g of fiber per 100 g on a wet basis, as shown in Table 1. According to our analyses, sorghum bread samples presented 5.79 ± 0.03; 4.71 ± 0.13; and 5.48 ± 0.03 g per 100 g of serving for brown, bronze, and white sorghum, respectively. According to the FDA [44], food is a good source of fiber if it contains 10 to 19% of the dietary reference intake (DRI) per the amount customarily consumed, and is high or rich in fiber if it has more than 20%. So, food with at least 2.8 g of fiber is considered high or rich in fiber if it presents more than 5.5 g of dietary fiber per serving, since the fiber recommendation is 25 to 28 g per day. Thus, our brown sorghum bread can be classified as rich in fiber, and the other samples are a good fiber source.

Since fiber can be defined as any non-digestible carbohydrate and lignin not degraded in the upper gut, it has essential roles in decreasing postprandial glucose response [44] . Fiber is also associated with a reduced risk of cardiovascular diseases and diabetes mellitus type 2 (DM2), besides its relevance in DM2 treatment, lowering blood cholesterol, increasing satiety, and preventing constipation [45]. Therefore, the consumption of our sorghum bread samples, which are good fiber sources and rich in fiber, can improve human health due to several previously described benefits.

Another factor that influences postprandial glucose response is the presence of tannins. According to Poquette et al. [29], tannins in sorghum contribute to the poor digestibility of starch, which may lead to the slow absorption of carbohydrates. The lowest starch digestibility of sorghum among cereals is due to the strong association among starch, proteins, and tannins [7]. Additionally, sorghum proteins, especially after cooking, present a lower digestibility than cereals such as wheat and maize [46]. Furthermore, according to Espetia-Hernández [47], tannins form complexes with proteins and iron, reducing sorghum digestibility. Therefore, there are several mechanisms that may explain the low GI observed in sorghum bread, an added advantage for the glycemic control of type 2 diabetic people.

Furthermore, brown sorghum bread presented the highest antioxidant activity value than other samples (ORAC assay: 45.49 μmol TE/g). This result was probably because colored sorghum genotypes, as brown and red, had higher phenolic compounds concentration [16]. On the contrary, white sorghum presented the lowest antioxidant activity (ORAC assay: 22.41 μmol TE/g), similar to the study conducted by Awika and Rooney [8], in which white sorghum grain presented only 22 μmol TE/g. These findings can be explained because white sorghum is usually rich in tannins and has a reduced antioxidant activity. Moreover, according to Le Bourvellec and Renard [48], sorghum condensed tannins can bind starch and polysaccharides. Since phenolic compounds can form complexes with proteins and carbohydrates in foods leading to changes in structural properties that impact digestibility, this process is related to a decrease in glucose response.

Moraes et al. [15] found that the estimated glycemic index of sorghum flour was negatively correlated to phenolic compounds, specific flavonoids, antioxidant activity, and total, insoluble and soluble dietary fiber and b-glucan. However, RS did not correlate to the estimated GI. Basu et al. [49] studied the glucose metabolism with healthy volunteers after either rice (*n* = 8) or sorghum (*n* = 8) mixed meals consumption, and reported higher insulin sensitivity with sorghum than rice meals (identical calorie and macronutrient compositions).

Based on this result, we can speculate that sorghum bread’s low postprandial glucose response may occur due to an improved insulin sensitivity without reducing insulin release. Galarregui et al. [50] presented that subjects with higher values of antioxidant capacity had a significantly lower insulin resistance (HOMA-IR) and correlation analyses showed inverse associations between GI and antioxidant capacity. In addition, a study conducted by Rosén et al. [51], with different varieties of rye bread, found that the content of phenolic compounds was negatively related to the early glucose response (T 0–60 min). The mechanism is probably multifactorial, including the effects of dietary fiber and a lowered rate of starch hydrolysis. Therefore, more studies are important to explore this complex mechanism.

Another study with sorghum, conducted by Park et al. [52], concluded that the administration of sorghum extract in mice significantly reduced serum glucose levels. However, only the treatment with a higher concentration significantly lowered the serum insulin level. Accordingly, Lakshmi et al. [53] demonstrated that the consumption of whole sorghum significantly lowered fasting and glucose 2 h AUC in type 2 diabetic individuals, likely due to fiber content.

Although dietary fiber lowers blood glucose levels by delaying gastric emptying, intestinal transit time, and carbohydrate absorption, Ray et al. [54] reported that consuming sorghum grain did not affect serum glucose or insulin levels compared with those in hyperlipidemic rats fed white rice [55]. These inconsistent results can be related to the type of sorghum consumed, sorghum grain versus extract, or the animal model used [52]. Therefore, the mechanisms of sorghum consumption and their effect on glucose, insulin, and antioxidant responses are still unclear and need more research.

This research had limitations due to the few sorghum genotypes (white, brown, and bronze) analyzed in gluten-free bread. Additionally, there was no direct gluten determination. Nevertheless, it is important to highlight that a study conducted with different sample profiles and other sorghum genotypes and products, besides bread, would be recommended since the interaction of ingredients may occur and present differences in GI.

## 5. Conclusions

Our results indicated that brown sorghum bread was the only sample classified as low GI and presented an improvement in postprandial glycemic responses in healthy adult men. This finding may occur due to a significantly higher fiber amount and the antioxidant activity of brown sorghum bread. Therefore, the consumption of brown sorghum should be encouraged as it produced better GFP due to its nutritional profile and health benefits than rice bread did. Finally, more research is required to explore the effects of different sorghum genotypes in food products on human health with different populations, including celiacs.

## Figures and Tables

**Figure 1 foods-10-02256-f001:**
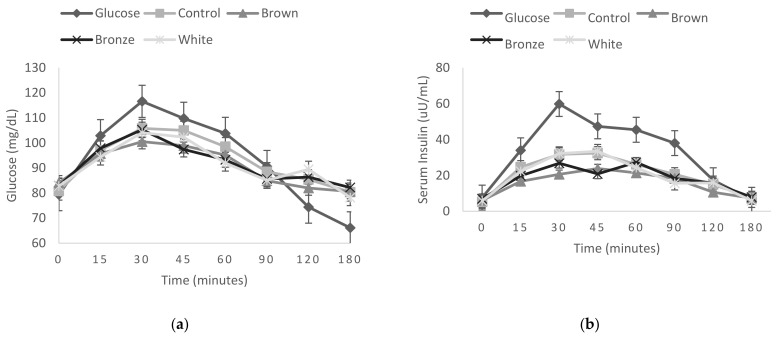
Fasting and postprandial glycemic (**a**) and insulinemic (**b**) responses to test meals bread intake containing three different sorghum genotypes and the control (rice bread). Values presented by means and their standard errors are represented by vertical bars.

**Table 1 foods-10-02256-t001:** Chemical composition (g/100 g) and antioxidant activity of control and sorghum bread.

	Control	Brown	Bronze	White
Moisture	40.97 ± 0.73 ^a^	47.42 ± 0.91 ^b^	46.94 ± 0.74 ^b^	48.80 ± 0.46 ^b^
Ashes	1.51 ± 0.00 ^b^	1.53 ± 0.00 ^b^	1.52 ± 0.02 ^b^	1.35 ± 0.00 ^a^
Carbohydrate	37.51 ± 0.85 ^c^	31.68 ± 0.54 ^b^	29.29 ± 0.56 ^a^	30.60 ± 0.39 ^ab^
Resistant starch	3.05 ± 0.05 ^b^	1.77 ± 0.12 ^a^	2.75 ± 0.19 ^b^	1.55 ± 0.06 ^a^
Fiber	3.96 ± 0.03 ^a^	5.79 ± 0.03 ^d^	4.71 ± 0.13 ^b^	5.48 ± 0.03 ^c^
Protein	5.36 ± 0.52 ^a^	5.42 ± 0.24 ^a^	6.13 ± 0.15 ^b^	5.36 ± 0.18 ^a^
Lipids	7.58 ± 0.00 ^a^	6.41 ± 0.08 ^b^	8.68 ± 0.00 ^d^	6.87 ± 0.00 ^c^
ORAC (μmol TE/g)	25.60 ± 2.77 ^a^	45.49 ± 2.07 ^b^	30.84 ± 0.28 ^a^	22.41 ± 3.04 ^a^

Values in the same line marked with different letters show statistical significance (*p* < 0.05).

**Table 2 foods-10-02256-t002:** Glycemic and insulinemic index (%) of control and sorghum bread.

	Glucose 3 h AUC	Glycemic Index	Insulin 3 h AUC	Insulin Index
Glucose	2619.75 ± 2094.94 ^a^	100	4797.29 ± 3009.89 ^a^	100
Control	2098.50 ± 1352.53 ^a^	80	3372.05 ± 3255.73 ^b^	70
Brown	1144.50 ± 590.67 ^b^	44	2379.59 ± 3083.12 ^b^	50
Bronze	1571.25 ± 908.22 ^ab^	60	2697.02 ± 2890.74 ^b^	56
White	1662.75 ± 1362.39 ^ab^	63	2094.09 ± 1212.01 ^b^	44

Values in the same column marked with different letters show statistical significance (*p* < 0.05).

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
