# Peer review of "Impact of Gluten-Free Sorghum Bread Genotypes on Glycemic and Antioxidant Responses in Healthy Adults"

_foods, 2021, doi:10.3390/foods10102256_

Round 1

Reviewer 1 Report

I have attached my comments in a document. 

Author Response

Response to Editor and Reviewers

 We are sending the responses as requested by reviewers. The authors appreciate the opportunity of considering our work and also the valuable suggestions to improve its presentation. We want to thank you for your time in conducting the review since, with your contributions, the article has improved substantially. All corrections were handled in the best possible way and with great care. We hope that the revised version is suitable for publication in this important journal.

 Reviewer 1 - Comments and Suggestions for Authors

The manuscript investigates the use of different sorghum genotypes in GF formulations for better metabolic responses like GI. The language of the manuscript is well even though there are some minor issues. The authors did not provide a detailed discussion explaining the reasons behind the differences observed. Some part of the Discussion was repetition of the Results. The authors should provide more in-depth discussion and explain the mechanisms or reasons with scientific findings. Especially, the high variations in glucose and insulin levels make the findings questionable, due to which I am suggesting having more participants in the study for robust results. In the light of these, I recommend a major revision for this manuscript. In addition to my general comments, my other comments are listed below for each section.

R: Thank you for your comments and contributions. We improved the discussion as suggested. All the other comments are listed below as requested.

Abstract

L 17 – resistant starch and dietary fiber content

R: Changed accordingly.

L 18 – responses

R: Changed accordingly.

L 19-21 – It does not make sense to have the sentence about ANOVA in the abstract. Please consider deleting.

R: Changed accordingly.

L 24 – than “those of the” control

R: Changed accordingly.

L 25 – Please consider rewriting this sentence to show the superiority of brown sorghum and its possible implications in the gluten-free market.

R: Changed accordingly (lines 25-27).

Introduction

L 35 – delete non-gluten content

R: Changed accordingly.

L 35-36 – change as “incorporated into gluten-free bread formulations as well as other baked …”

R: Changed accordingly.

L 38 – change color with genotype – each genotype

R: Changed accordingly.

L 39-41 – change the information explained with color difference to genotype information according to the comment above

R: Changed accordingly.

L 46-47 – Please merge these paragraphs

R: Changed accordingly.

L 54 – Reference is needed at the end of this sentence

R: We deleted the information since it is not that important to the context.

L 57 – change “human’s results” to “research in human” or “clinical studies”

R: Changed accordingly.

Please consider expanding the objective paragraph by linking the gap in the literature to the importance of the use of sorghum in the GF market.

R: Changed accordingly (lines 80 and 81).

Results

Table 2 – Glucose and insulin 3h AUC values showed huge variations as shown with high standard deviation. How did the authors explain this? Even though a statistical analysis was conducted, the variation does not make sense to explain the data collected. These huge 2 variations indicate that either more participants should have included in the study or the data of participants showing extreme differences from others should have been removed.

R: One participant was removed due to abnormal basal (>20 µU/mL) and insulin response curve. In addition, the sample size calculation was performed, resulting in nine individuals (crossover design), and we conducted the experiments with ten participants. Finally, the laboratory activity was suspended because of the pandemic scenario, precluding increasing the sample size. Unfortunately, we observed high variability of data which eventually occurs in human clinical trials studies.

 Most of the Discussion only contains the results and there is no discussion explaining the differences between the samples. Basically, the results were repeated (e.g., the paragraph in L 285-292).

R: Several changes were made to improve discussion (lines 349-351, 354-359, 363-365, 368-371).

 L 268-269 – merge the paragraphs

R: Changed accordingly.

 L 274-284 – It does not make sense to compare the samples prepared in this study with a commercial bread as you mentioned the ingredients and their compositions were completely different. I do not understand the reason why such a comparison was made.

R: Changed accordingly; paragraph was removed.

 L 295 – If low GI improves some metabolic activities as indicated in this line, how do the authors explain the profile differences in glycemic index and insulin index after the consumption of bread made with different sorghum genotypes? For example, as GI of brown sorghum bread was the lowest, it is not the case for II index.

R: We can speculate that sorghum bread's low postprandial glucose response may occur due to improved insulin sensitivity without reducing insulin release. One sentence was added in the discussion to make this clear to the readers (lines 395-397).

 L 308-313 – Please consider rewriting this paragraph. I understand that the authors want to promote the GF products made with sorghum by providing the fact that its high fiber content resulting lower GI. However, the written version of this paragraph does not give that to readers and it needs improvement.

R: Changed accordingly (lines 309-311).

 L 330 – This paragraph only includes general knowledge and does not contribute to the Discussion. It should be written as an implication if wanted to be included.

R: Changed accordingly (lines 350-352).

L335 – In which way the tannins are affecting the starch digestibility. Is it because they are modifying starch’s structure or eliminating the starch digestion enzymes? Or is there any other explanation behind it? Please elaborate with some discussion and explain the possible mechanism.

R: Several mechanisms can explain the lowest digestibility of sorghum and are presented in lines 355 to 360.

 L 340-344 – Again only results were repeated, but there is no discussion.

R: Changes were made to provide discussion (lines 364-366, 369-372).

 Thank you for your comments!

Reviewer 2 Report

The problem undertaken at work („Impact of gluten-free sorghum bread genotypes on glycemic and antioxidant responses in healthy adults”) is interesting, however, in the manuscript, there are some places that must be revised. Title, keywords, abstracts are clearly describes what the manuscript is about. There is introduction to justify that the work is included in the scope of the journal "Foods”. Materials and Methods should be supplemented. Data of experiments are not always properly analyzed and interpreted.

Detailed comments

Regarding Introduction

All abbreviations should be explained, e.g. L.103 : „RS” or “GF”

L.52-54: “Meanwhile, for anthocyanins, brown sorghum presents 1.6 to 3.9 mg/g, black sorghum 4.0 to 9.8 mg/g, and red sorghum 3.3 mg/g of anthocyanins.” - information on literature sources is missing.

L.56-57: “Despite phenolic compounds’ bioavailability after dietary intake being a research topic in recent years, human’s results are scarce and controversial [7,19].” - What is the controversy? Please explain.

Regarding “2.1. Sorghum bread production”

This paragraph describes the production of Sorghum bread and raw materials - the title needs to be changed.

L.78-80: “Four bread samples were prepared…. with the different types of sorghum flours (white, brown, and bronze), according to Andrade de Aguiar et al. [21].” - This sentence is not clear. Moreover, the information on raw materials are needed, e.g. manufacturer, type.

L.86-93: Was the samples of bread prepared according Andrade de Aguiar et al. [21]?

In this chapter many information is missing, for example: mass of dough in a cake tin; tape of oven, manufacturer, country;  material of plastic bags, storage time, how long time were the samples frozen. Moreover why was the baking time of the samples different?

RegardingFood analyses”

This part needs to be completed, e.g .:

producent, kraju produkcji aparatów, (e.g.: Extractor Ankom Model XT10, a shaking water bath, centrifuge, incubator, magnetic stirrer).

What spectrophotometer was used?

What was the reagent blank?

Titles: „2.4. Participants” and „2.5. Study design” should be changed and described

Regarding „3.4. Antioxidant capacity.”- Please show where the results are?

Regarding "Table 1 – Please correct statistic for example : „a” should be always with the littlest value.

Regarding „Results” and „Discusion”,

Chapters: „Results” and „Discusion”, should be corrected (phrase repeating, should be eliminated)

Discussion should be shortened to discussion of own results only with significant literature information  e.g. L.265-320.

Author Response

Response to Editor and Reviewers

We are sending the responses as requested by reviewers. The authors appreciate the opportunity of considering our work and also the valuable suggestions to improve its presentation. We want to thank you for your time in conducting the review since, with your contributions, the article has improved substantially. All corrections were handled in the best possible way and with great care. We hope that the revised version is suitable for publication in this important journal.

Reviewer 2 - Comments and Suggestions for Authors

The problem undertaken at work („Impact of gluten-free sorghum bread genotypes on glycemic and antioxidant responses in healthy adults”) is interesting, however, in the manuscript, there are some places that must be revised. Title, keywords, abstracts are clearly describes what the manuscript is about. There is introduction to justify that the work is included in the scope of the journal "Foods”. Materials and Methods should be supplemented. Data of experiments are not always properly analyzed and interpreted.

Regarding Introduction

All abbreviations should be explained, e.g. L.103 : „RS” or “GF”

R: They were explained in the introduction section.

L.52-54: “Meanwhile, for anthocyanins, brown sorghum presents 1.6 to 3.9 mg/g, black sorghum 4.0 to 9.8 mg/g, and red sorghum 3.3 mg/g of anthocyanins.” - information on literature sources is missing.

R: As requested by reviewer 1, this line was removed.

 L.56-57: “Despite phenolic compounds’ bioavailability after dietary intake being a research topic in recent years, human’s results are scarce and controversial [7,19].” - What is the controversy? Please explain.

R: A paragraph was added to explain the controversies (lines 60-65).

Regarding “2.1. Sorghum bread production”

This paragraph describes the production of Sorghum bread and raw materials - the title needs to be changed.

R: Changed accordingly.

 L.78-80: “Four bread samples were prepared…. with the different types of sorghum flours (white, brown, and bronze), according to Andrade de Aguiar et al. [21].” - This sentence is not clear. Moreover, the information on raw materials are needed, e.g. manufacturer, type.

R: Changed accordingly (lines 84-86, 90, and 91). We used the same recipe described by Andrade de Aguiar et al., but we gave more details and explained the procedures to prepare the bread samples.

L.86-93: Was the samples of bread prepared according Andrade de Aguiar et al. [21]?

R: Yes, but we added information about procedures.

In this chapter many information is missing, for example: mass of dough in a cake tin; tape of oven, manufacturer, country;  material of plastic bags, storage time, how long time were the samples frozen. Moreover why was the baking time of the samples different?

R: Changed accordingly (lines 103, 105-109). The baking time was different because the doughs’ textures needed more time for sorghum bread samples. For each cake tin, we placed 447g of dough before baking.

 Regarding “Food analyses”

This part needs to be completed, e.g .:

producent, kraju produkcji aparatów, (e.g.: Extractor Ankom Model XT10, a shaking water bath, centrifuge, incubator, magnetic stirrer).

R: Changed accordingly (lines 119, 121, 125, 132, 205, and 206).

What spectrophotometer was used?

R: It was used the Biochrom (line 132).

 What was the reagent blank?

R: The reagent blank was 0.1 mL of sodium acetate buffer with pH 4.5 + 3 mL of GOPOD (lines 132 and 133).

 Titles: „2.4. Participants” and „2.5. Study design” should be changed and described

R: Changed accordingly.

 Regarding „3.4. Antioxidant capacity.”- Please show where the results are?

R: The results are described in lines 276-278.

 Regarding "Table 1 – Please correct statistic for example : „a” should be always with the littlest value.

R: Changed accordingly.

 Regarding „Results” and „Discusion”,

Chapters: „Results” and „Discusion”, should be corrected (phrase repeating, should be eliminated)

R: Several changes were made to avoid repetition.

 Discussion should be shortened to discussion of own results only with significant literature information  e.g. L.265-320.

R: Several changes were made. Reviewers 1 and 3 asked for more discussion regarding the lowest digestibility of sorghum that was included.

 Thank you for your comments!

Reviewer 3 Report

Dear Authors,

I revised the manuscript “Impact of gluten-free sorghum bread genotypes on glycemic and antioxidant responses in healthy adults” with pleasure.

The paper is well written. The Introduction guides the reader to the aim of the paper. Methods are sound. Results are clearly reported and properly discussed.

I suggest addressing the following comments:

  • Lines 50-54: I suggest removing this sentence from the Introduction. In my opinion, it is not necessary to explain the rationale and the aim of the study.
  • Line 83: please, replace “gun” with “gum”.
  • Line 96: I suggest amending the title as “Bread Chemical Composition”.
  • Lines 151: why was the study performed on 13 subjects? In lines 136, a sample size of 9 participants was reported. How many male subjects? How many female subjects?
  • Lines 314: there are several studies on GFP quality. Please, consider also:
    • Fry, L.; Madden, A.M.; Fallaize, R. An investigation into the nutritional composition and cost of gluten-free versus regular food products in the UK. J. Hum. Nutr. Diet. 2018, 31, 108–120.
    • Allen, B.; Orfila, C. The Availability and Nutritional Adequacy of Gluten-Free Bread and Pasta. Nutrients 2018, 10, 1370.
    • Melini, V.; Melini, F. Gluten-Free Diet: Gaps and Needs for a Healthier Diet. Nutrients 2019, 11, 170.
    • Cornicelli, M.; Saba, M.; Machello, N.; Silano, M.; Neuhold, S. Nutritional composition of gluten-free food versus regular food sold in the Italian market. Dig. Liver Dis. 2018, 50, 1305–1308.

Author Response

We are sending the responses as requested by reviewers. The authors appreciate the opportunity of considering our work and also the valuable suggestions to improve its presentation. We want to thank you for your time in conducting the review since, with your contributions, the article has improved substantially. All corrections were handled in the best possible way and with great care. We hope that the revised version is suitable for publication in this important journal.

Reviewer 3 - Comments and Suggestions for Authors

I revised the manuscript “Impact of gluten-free sorghum bread genotypes on glycemic and antioxidant responses in healthy adults” with pleasure.

The paper is well written. The Introduction guides the reader to the aim of the paper. Methods are sound. Results are clearly reported and properly discussed.

R: Thank you!

 Lines 50-54: I suggest removing this sentence from the Introduction. In my opinion, it is not necessary to explain the rationale and the aim of the study.

R: Changed accordingly. The last sentence of the paragraph was removed.

 Line 83: please, replace “gun” with “gum”.

R: Changed accordingly.

Line 96: I suggest amending the title as “Bread Chemical Composition”.

R: Changed accordingly.

Lines 151: why was the study performed on 13 subjects? In lines 136, a sample size of 9 participants was reported. How many male subjects? How many female subjects?

R: We included in this line that all 13 participants were male.

A minimum sample of 9 participants was necessary for the study. It is stated in line 136 as the statistical requirement for the study. However, we started the study with 13 since participants could not come or refuse to continue throughout the study. Line 151 described that we started the study with 13 male individuals.

Lines 314: there are several studies on GFP quality. Please, consider also:

  •  Fry, L.; Madden, A.M.; Fallaize, R. An investigation into the nutritional composition and cost of gluten-free versus regular food products in the UK. J. Hum. Nutr. Diet. 2018, 31, 108–120.
  • Allen, B.; Orfila, C. The Availability and Nutritional Adequacy of Gluten-Free Bread and Pasta. Nutrients 2018, 10, 1370.
  • Melini, V.; Melini, F. Gluten-Free Diet: Gaps and Needs for a Healthier Diet. Nutrients 2019, 11, 170.
  • Cornicelli, M.; Saba, M.; Machello, N.; Silano, M.; Neuhold, S. Nutritional composition of gluten-free food versus regular food sold in the Italian market. Dig. Liver Dis. 2018, 50, 1305–1308.

R: Thank you for your suggestions. We included more discussion regarding articles that we added, including lines 333 to 337.

 Thank you for your comments!

Round 2

Reviewer 2 Report

Dear Authors

The manuscript is carefully prepared and is suitable for publication in Foods in its current form.